# MULTIVARIATE TIME SERIES FORECASTING WITH LATENT GRAPH INFERENCE

## ABSTRACT

This paper introduces a new architecture for multivariate time series forecasting that simultaneously infers and leverages relations among time series. We cast our method as a modular extension to univariate architectures where relations among individual time series are dynamically inferred in the latent space obtained after encoding the whole input signal. Our approach is flexible enough to scale gracefully according to the needs of the forecasting task under consideration. In its most straight-forward and general version, we infer a potentially fully connected graph to model the interactions between time series, which allows us to obtain competitive forecast accuracy compared with the state-of-the-art in graph neural networks for forecasting. In addition, whereas previous latent graph inference methods scale $O(N^2)$ w.r.t. the number of nodes $N$ (representing the time series), we show how to configure our approach to cater for the scale of modern time series panels. By assuming the inferred graph to be bipartite where one partition consists of the original $N$ nodes and we introduce $K$ nodes (taking inspiration from low-rank-decompositions) we reduce the time complexity of our procedure to $O(NK)$. This allows us to leverage the dependency structure with a small trade-off in forecasting accuracy. We demonstrate the effectiveness of our method for a variety of datasets where it performs better or very competitively to previous methods under both the fully connected and bipartite assumptions.

## 1 INTRODUCTION

Time Series Forecasting (TSF) has been widely studied due to its practical significance in a wide variety of applications such as climate modelling (Mudelsee, 2019), supply chain management in retail (Larson, 2001; Böse et al., 2017), market analysis in finance (Andersen et al., 2005), traffic control (Li et al., 2017) and medicine (Kaushik et al., 2020). Petropoulos et al. (2020) provide a non-systematic overview of further applications. In TSF, given a sequence of data points indexed over time, we aim to estimate its future values based on previously observed data. Data is often multivariate, meaning that multiple variables vary over time, each variable may not only depend on its own historical values, but also on other variables' past. Efficiently modelling the dependencies between these variables is still an open problem.

Multivariate Time Series (MTS) methods aim to leverage the dependencies between variables in order to improve the forecasting accuracy. Some classical MTS forecasting algorithms such as Vector Autoregression (VAR) (Lütkepohl, 2005) or Gaussian Processes (Roberts et al., 2013) only consider linear dependencies among variables. A natural way to model non-linear dependencies in deep learning is via Graph Neural Networks (GNNs) (Bruna et al., 2013; Defferrard et al., 2016; Kipf and Welling, 2016). In fact, GNNs have been successfully applied for the Multivariate Time Series forecasting (Li et al., 2017; Yu et al., 2017; Chen et al., 2020a), leveraging the relations among different series. But these methods require a pre-defined adjacency matrix which may only be available in some specific datasets, for example, traffic datasets, where it can be constructed from the spatial structure of a city. More recently, a family of methods that do not require a pre-defined adjacency matrix have been proposed (Wu et al., 2019; Franceschi et al., 2019; Wu et al., 2020; Shang et al., 2021). In this case, a latent graph representation is inferred while forecasting, allowing to operate on a larger variety of datasets. Our work belongs to this category.

However, inferring all pairwise relations may come at a higher computational cost, as the number of relations scales quadratically $O(N^2)$ w.r.t. the number of nodes. This limits the scalability to large datasets. Additionally, the above mentioned latent graph inference methods perform message passing update at every time step iteration which can also be expensive.

To overcome these limitations, we propose a new latent graph inference algorithm for Multivariate Time Series forecasting that is more efficient than previous algorithms while achieving better or competitive performance. We cast the latent graph inference as a modular and easy-to-implement extension to current univariate models. The graph is dynamically inferred per inputted data stream allowing a more flexible representation than a static graph for the whole dataset. Additionally, we optionally reduce the complexity from $O(N^2)$ (Fully Connected Assumption) to $O(NK)$ (Bipartite Assumption) where $K \ll N$ for a small trade off in performance.

## 2 BACKGROUND

### 2.1 TIME SERIES FORECASTING

In time series forecasting we want to estimate a future time series $\mathbf{x}_{t+1:T}$ given its past $\mathbf{x}_{t_0:t}$ where $t_0 \leq t \leq T$ indexes over time, and (optionally) some context information $\mathbf{c}$. For the multivariate case we assume the time series is composed of $N$ variates at a time such that $\mathbf{x}_{t_0:T} = \{\mathbf{x}_{1,t_0:T}, \ldots, \mathbf{x}_{N,t_0:T}\} \in \mathbb{R}^{N \times T - t_0 + 1}$. In this section we distinguish two main categories of time series forecasting methods, Global Univariate and Multivariate.

**Global Univariate methods**: In this case we only use the past of each univariate to predict its future. However, the model weights $\theta_u$ are shared across all univariate time series. More formally:

$$\hat{\mathbf{x}}_{i,t+1:T} = f_u(\mathbf{x}_{i,t_0:t}, \mathbf{c}_i; \theta_u) \tag{1}$$

where $\hat{\mathbf{x}}$ denotes the estimated values, $i \in \{1, \ldots, N\}$ indexes over multivariates and $f_u(\cdot)$ is the estimator function with learnable parameters $\theta_u$ shared across time series. Conditioning on the past of each univariate may limit the performance of the forecasting algorithm compared to multivariate ones. Despite that, it simplifies the design of $f_\theta$ and already provides reasonable results. A popular example of univariate models would be (Salinas et al., 2020).

**Multivariate methods**: Multivariate methods condition on all past data (all $N$ variates) and directly predict the multivariate target. More formally:

$$\hat{\mathbf{x}}_{t+1:T} = f_m(\mathbf{x}_{t_0:t}, \mathbf{c}). \tag{2}$$

Different variables may be correlated and/or depend on the same con-founders. For example, in retail forecasting, PPE masks and antibacterial soaps jointly increased in demand during the early days of the COVID-19 pandemic. In traffic forecasting, an increase of the outcome traffic flow in a given neighborhood may result in an increase of the income traffic flow on another one. Modelling these dependencies may improve the forecasting accuracy, but it may come at a cost of higher complexity and hence more expensive algorithms, specially when trying to model all pairwise interactions between variates.

### 2.2 GRAPH NEURAL NETWORKS

Graph Neural Networks (GNNs) (Bruna et al., 2013; Defferrard et al., 2016; Kipf and Welling, 2016) operate directly on graph structured data. They have gained a lot of attention in the last years due to their success in a large variety of domains which benefit from modelling interactions between different nodes/entities. In the context of multivariate time series, GNNs can be used to model the interactions between time series. In this work we consider the type of GNN introduced by (Gilmer et al., 2017). Given a graph $\mathcal{G} = (\mathcal{V}, \mathcal{E})$ with nodes $v_i \in \mathcal{V}$ and edges $e_{ij} \in \mathcal{E}$, we define a graph convolutional layer as:

$$\mathbf{m}_{ij} = \phi_e(\mathbf{h}_i^l, \mathbf{h}_j^l) \qquad \mathbf{m}_i = \sum_{j \in \mathcal{N}(i)} \alpha_{ij} \mathbf{m}_{ij} \qquad \mathbf{h}_i^{l+1} = \phi_h(\mathbf{h}_i^l, \mathbf{m}_i) \tag{3}$$

Where $\phi_e$ and $\phi_h$ are the edge and node functions, usually approximated as Multi Layer Perceptrons (MLPs), $\mathbf{h}_i^l \in \mathbb{R}^{\text{nf}}$ is the nf-dimensional embedding of a node $v_i$ at layer $l$ and $\mathbf{m}_{ij}$ is the edge

embedding that propagates information from node $v_j$ to $v_i$. A GNN is constructed by stacking multiple of these Graph Convolutional Layers $\mathbf{h}^{l+1} = \text{GCL}[\mathbf{h}^l, \mathcal{E}]$. Additionally, in Equation 3 we include $\alpha_{ij} \in (0,1)$ which is a scalar value that performs the edge inference or attention over the neighbors similarly to Veličković et al. (2017). As done in (Satorras et al., 2021), we choose this value to be computed as the output of a function $\alpha_{ij} = \phi_\alpha(\mathbf{m}_{ij})$ where $\phi_\alpha$ is composed of just a linear layer followed by a sigmoid activation function.

## 3 RELATED WORK

Time series forecasting has been extensively studied in the past due to its practical significance with a number of recent overview articles available (et al., 2020; Benidis et al., 2020; Lim and Zohren, 2021). Traditionally, most classical methods are univariate in nature (see e.g., Hyndman and Athanasopoulos (2017) for an overview). While some of these have multi-variate extensions (e.g., ARMA and VARMA models), they are limited by the amount of related time series information they can incorporate. Dynamic factor models (Geweke, 1977; Wang et al., 2019a) are fore-runners of a family of models that has recently received more attention, the so-called global models (Januschowski et al., 2019; Montero-Manso and Hyndman, 2022). These global models estimate their parameters over an entire panel of time series, so thereby taking advantage of cross time series learning, but still produce a univariate forecast. Many such global models have been proposed building on the main neural network architectures like RNNs (Salinas et al., 2020; Liberty et al., 2020; Bandara et al., 2019), CNNs (Wen et al., 2017; Chen et al., 2020b), Transformers (Li et al., 2019; Lim et al., 2021; Eisenach et al., 2020) and also combining classical probabilistic models with deep learning (Rangapuram et al., 2018; Kurle et al., 2020; de Bézenac et al., 2020). However, these global models do not explicitly model the relationship between the time series in the panel.

Most recently, global multi-variate forecasting models have received attention, in particular models that attempt to capture the relationship of the time series via a multi-variate likelihood (Rasul et al., 2020; 2021; de Bézenac et al., 2020; Salinas et al., 2019). Here, we attempt to capture the multi-variate nature of many modern forecasting problems primarily by using a multi-variate time series as input. For this, a natural way to model and exploit the relationship between time series is via Graph Neural Networks (Bruna et al., 2013; Defferrard et al., 2016; Kipf and Welling, 2016) which have been successfully applied to a wide variety of deep learning domains where exploiting relations between entities/nodes can benefit the prediction task. Even in those cases where edges are not explicitly provided in the dataset, attention or a latent graph can be inferred from the node embeddings such that GNNs can still leverage the structure of the data. Some examples of latent graph inference or attention are (Wang et al., 2019b) in point clouds, (Franceschi et al., 2019) in semi-supervised graph classification, (Ying et al., 2018) in hierarchical graph representation learning, (Kipf et al., 2018) in modelling dynamical systems, (Kazi et al., 2020) in zero-shot learning and 3D point cloud segmentation, (Garcia and Bruna, 2017; Kossen et al., 2021) in image classification, (Cranmer et al., 2020) in inferring symbolic representations and (Fuchs et al., 2020; Satorras et al., 2021) in molecular property prediction.

In Multivariate Time Series (MTS) forecasting we can leverage dependencies between time series by exchanging information among them. Lai et al. (2018); Shih et al. (2019) are some of the first deep learning approaches designed to exploit those pair-wise dependencies. More recent methods, (Li et al., 2017; Yu et al., 2017; Seo et al., 2018; Zhao et al., 2019) are built in the intersection of Graph Neural Networks and time series forecasting but they require a pre-defined adjacency matrix. Lately, new methods that infer a latent graph from the node embeddings have been introduced in MTS forecasting (Wu et al., 2020; Shang et al., 2021; Cao et al., 2021), and in MTS anomaly detection (Zhang et al., 2020; Deng and Hooi, 2021). These methods can be applied to any dataset even when there is not an explicitly defined adjacency matrix. But this comes with a limitation, inferring the edges can be expensive since those scale quadratically $O(N^2)$ w.r.t the number of nodes/variables $N$ or $O(N^3)$ in (Cao et al., 2021).

Our approach is related to (Wu et al., 2020; Shang et al., 2021), but in contrast a) Our latent graph is dynamically inferred for each inputted data stream instead of a static graph for the whole dataset which allows a more flexible graph representation. b) It is modular since it can be added as an extension to standard univariate methods after encoding the input signal $\mathbf{x}_{t_0:t}$, this also makes the

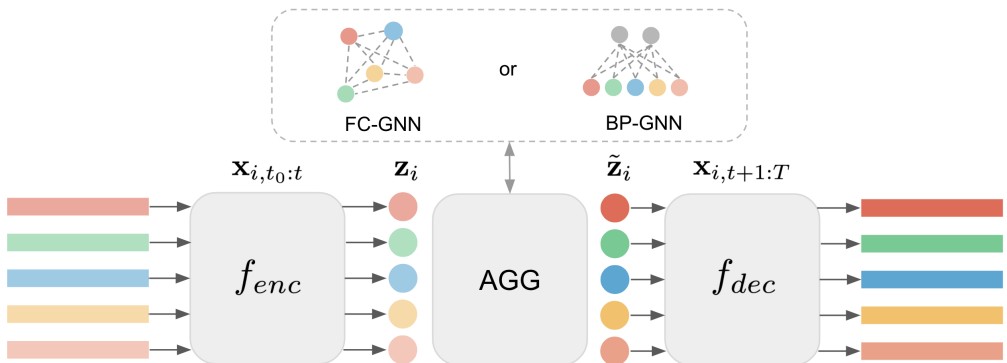

Figure 1: Illustration of the presented method under fully connected and bipartite graph assumptions.

graph operation cheaper since the message exchange is only done at time step $t$. c) Optionally reduces the number of edges from $O(N^2)$ to $O(NK)$ where $K \ll N$ with the bipartite assumption.

## 4 METHOD

In Section 2.1 we discussed two different sorts of forecasting methods: Global Univariate Models and Multivariate Models. We cast our multivariate algorithm as a modular extension of the univariate case from Equation 1. We can break down the Univariate Model in two main steps $f_u = f_{enc} \circ f_{dec}$ such that $\mathbf{x}_{i,t_0:t} \xrightarrow{f_{enc}} \mathbf{z}_i \xrightarrow{f_{dec}} \hat{\mathbf{x}}_{i,t+1:T}$, where $f_{enc}$ encodes the input signal $\mathbf{x}_{i,t_0,t}$ (and optionally some context information $\mathbf{c}_i$) into an embedding $\mathbf{z}_i$ and $f_{dec}$ estimates the future signal from this embedding. In our method, we include a multivariate aggregation module AGG in between $f_{enc}$ and $f_{dec}$ that propagates information among nodes in the latent space $\mathbf{z} = \{\mathbf{z}_1, \ldots, \mathbf{z}_N\}$. This aggregation module takes as input the embedding $\mathbf{z}_i = f_{enc}(\mathbf{x}_{i,t_0:t}, \mathbf{c}_i)$, and outputs a modified embedding $\hat{\mathbf{z}} = \text{AGG}(\mathbf{z})$ where information has been propagated among nodes. These new embeddings are then passed as input to the decoder $f_{dec}$. The resulting algorithm is:

$$\text{Univariate Encoder} \quad \mathbf{z}_i = f_{enc}(\mathbf{x}_{i,t_0:t}, \mathbf{c}_i) \tag{4}$$

$$\text{Multivariate extension} \quad \hat{\mathbf{z}} = \text{AGG}(\mathbf{z}) \tag{5}$$

$$\text{Univariate Decoder} \quad \hat{\mathbf{x}}_{i,t+1:T} = f_{dec}(\hat{\mathbf{z}}_i) \tag{6}$$

Notice the overall model is multivariate but $f_{enc}$ and $f_{dec}$ remain univariate, allowing a modular extension from current univariate methods to multivariate. Additionally, in contrast to the most recent methods (Wu et al., 2020; Shang et al., 2021), our model does not propagate information among nodes at every time step $[t_0, \ldots, t]$ but only in the AGG module. This makes the algorithm cheaper since the message propagation step is usually an expensive operation. Additionally, we experimented including a unique identifier of the time series in the encoded signal $\mathbf{z}_i$ as context information $\mathbf{c}_i = \text{id}$, which resulted in a significant improvement in accuracy as we will show in the experiments section. In the following subsections we propose two different Graph Neural Network solutions for the AGG module, each has its benefits and weaknesses regarding performance and scalability.

**Fully connected graph assumption | FC-GNN**

From the two assumptions, this is the most straightforward to implement given a standard GNN. It results in good performance, but its complexity scales $O(N^2)$ with respect to the number of nodes $N$. Despite this, it still resulted in faster computation times than previous $O(N^2)$ methods. Here, we directly use the Graph Neural Network defined in Section 2.2 as the aggregation module AGG. We assume a fully connected graph $\mathcal{G} = \{\mathcal{V}, \mathcal{E}\}$ where each time series embedding $\mathbf{z}_i$ is associated to a node of the graph $i \in \mathcal{V}$, specifically, the embedding $\mathbf{z}_i$ is provided as the input $\mathbf{h}_i^0$ to the GNN from Equation 3, then the GNN runs for $L$ layers and the output node embedding $\mathbf{h}_i^L$ is provided as the input $\hat{\mathbf{z}}_i$ of the decoder. Moreover, despite the fully connected assumption, the GNN infers attention weights $\alpha_{ij} \in (0, 1)$ (eq. 3) for each edge and input sample. This is equivalent to dynamically inferring a latent graph where edges are associated to soft values in the range $(0, 1)$. In the experiments section, we name this model a Fully Connected Graph Neural Network (FC-GNN). In summary, the

aggregation module AGG under this FC-GNN setting, is able to model non-linear relations among time series by exchanging pairwise messages $\mathbf{m}_{ij}$ (eq. 3) among them, these messages are non-linear embeddings of the edges and they are "gated" or multiplied by an inferred scalar value $\alpha_{ij}$.

**Bipartite graph assumption | BP-GNN**

In the previous section we presented a GNN method that exchanges information among time series under a fully connected graph assumption which computationally scales $O(N^2)$. In this section we introduce a bipartite graph assumption which reduces the complexity to $O(NK)$, $K$ being a parameter of choice $K \ll N$. To accomplish this, we define a bipartite graph $\mathcal{G} = (\mathcal{Y}, \mathcal{U}, \mathcal{E})$ with two sets of nodes $\mathcal{Y}$ and $\mathcal{U}$. Here $\mathcal{Y}$ is a set of $N$ nodes corresponding to the $N$ dimensions of the multivariate time series and $\mathcal{U}$ is a set of $K$ auxiliary nodes. Each time series embedding $\mathbf{z}_i$ is associated to one of the nodes $i \in \mathcal{Y}$. Additionally, we define embeddings $\mathbf{u} = \{\mathbf{u}_1, \ldots \mathbf{u}_K\}$ associated to the auxiliary set $\mathcal{U}$. Edges $\mathcal{E}$ interconnect all nodes between the two subsets $\{\mathcal{Y}, \mathcal{U}\}$, but there are no connections among nodes of the same subset. This results in $2NK$ edges, with a computation complexity of $O(NK)$.

The algorithm works in the following way. We input into the GNN the union of the two node subsets $\mathcal{V} = \mathcal{Y} \cup \mathcal{U}$. Specifically, the input embedding $\mathbf{h}^0$ defined in Equation 3 is the concatenation of the time series embeddings $\mathbf{z}$ with the auxiliary node embeddings $\mathbf{u}$ (i.e. $\mathbf{h}^0 = \mathbf{z} || \mathbf{u}$) where $\mathbf{u}$ are free learnable parameters initialized as Gaussian noise. Then, messages follow an asynchronous schedule, first they are sent from the time series nodes to the auxiliary nodes $\mathcal{Y} \to \mathcal{U}$, next the other way around $\mathcal{U} \to \mathcal{Y}$. Notice the auxiliary nodes $\mathcal{U}$ do not contain any information about the data until they have received messages from $\mathcal{Y}$, this is why we chose this schedule. A visual illustration of the whole algorithm is presented in Figure 1, where the bipartite graph is depicted at the top right of the image (BP-GNN).

We have conceptually defined the Bipartite Graph Neural Network. In Table 1 we introduce the equations that formally define it as an extension of the standard GNN Equation 3. Notice that it can be simply formulated as a two steps process where the indexes $i, j$ belong to each one of the subsets $\mathcal{U}$ or $\mathcal{V}$ depending on the direction of the messages ($\mathcal{V} \to \mathcal{U}$ or $\mathcal{U} \to \mathcal{V}$). Additionally, we used different learn-

| Step 1 $\mathcal{Y} \to \mathcal{U}$ | Step 2 $\mathcal{U} \to \mathcal{Y}$ |
|---|---|
| Equation 3 where $i \in \mathcal{U}$ & $j \in \mathcal{Y}$ | Equation 3 where $i \in \mathcal{Y}$ & $j \in \mathcal{U}$ |

Table 1: BP-GNN formulation.

able parameters between Step 1 and Step 2 in the modules $\phi_e$, $\phi_\alpha$ and $\phi_h$ since it resulted in better accuracy than sharing parameters. In Appendix A.1 we provide the BP-GNN equations from Table 1 in its full form. Following, we define the adjacency matrices corresponding to the two message passing steps (assuming all inference parameters $\alpha_{ij} = 1$):

$$A_1 = \begin{vmatrix} 0_{N \times N} & 0_{N \times K} \\ 1_{K \times N} & 0_{K \times K} \end{vmatrix}, \qquad A_2 = \begin{vmatrix} 0_{N \times N} & 1_{N \times K} \\ 0_{K \times N} & 0_{K \times K} \end{vmatrix}, \qquad \tilde{A} = A_2 A_1 = \begin{vmatrix} K_{N \times N} & 0_{N \times K} \\ 0_{K \times N} & 0_{K \times K} \end{vmatrix} \quad (7)$$

The two adjacency matrices $A_1$ and $A_2$ are non-symmetric and they define a directed message passing scheme. $A_1$ refers to $\mathcal{Y} \to \mathcal{U}$ and $A_2$ refers to $\mathcal{U} \to \mathcal{Y}$. After running these two propagation steps, all time series nodes $\mathcal{Y}$ will have received messages from all other time series nodes $\mathcal{Y}$ as in the fully connected graph case, but the number of messages have been reduced (iff $0 \leq K < N/2$) due to the factorization of the adjacency matrix. The first N rows and columns in matrix $\tilde{A}$ from Equation 7 represent the sum of all paths that communicate the time series nodes $\mathcal{Y}$ among them after the two updates $A_2 A_1$.

**Architecture details**

As explained in this Section 4, our method is composed of three main modules, the encoder $f_{enc}$, the decoder $f_{dec}$ and the aggregation function AGG. We choose to use relatively simple networks as encoder and decoder. The decoder $f_{dec}$ is defined as a Multi Layer Perceptron (MLP) with a single hidden layer and a residual connection in all experiments. The encoder $f_{enc}$ is also defined as an MLP for METR-LA, PEMS-BAY and our synthetic datasets and as a Convolutional Neural Network (CNN) for the other datasets since these require a larger encoding length and the translation equivariance of CNNs showed to be more beneficial. The encoder $f_{enc}$, first encodes the input signal $\mathbf{x}_{i,t_0:t}$ to an embedding vector by using the mentioned MLP or CNN, and then concatenates a unique

identifier ($\mathbf{c}_i = \text{id}$) to the obtained embedding vector resulting in $\mathbf{z}_i$. $\mathbf{c}_i$ could (optionally) include additional context information if it was provided in the dataset. The combination of these simple networks with our proposed aggregation module AGG fully defines our model. All architecture details are explained in detail in Appendix A.2.

The a aggregation module AGG was defined under two different assumptions in, the Fully Connected Graph assumption (FC-GNN) and the Bipartite Graph assumption (BG-GNN). In both cases the GNN is fully parametrized by the networks $\phi_e$, $\phi_h$ and $\phi_\alpha$. $\phi_e$ consists of a two layers MLP, $\phi_h$ is a one layer MLP with a skip connection from he input to the output and $\phi_\alpha$ is just a linear layer followed by a Sigmoid activation function. All these architecture choices are explained in more detail in Appendix A.2. In all experiments, the loss was computed as the Mean Squared Error between the estimated values $\hat{\mathbf{x}}_{t+1:T}$ from Equation 6 and the ground truth as $\mathcal{L} = l(\hat{\mathbf{x}}_{t+1:T}, \mathbf{x}_{t+1:T})$.

## 5 EXPERIMENTS

### 5.1 DATASETS AND BASELINES

We first evaluate our method in METR-LA and PEMS-BAY datasets from (Li et al., 2017) which record traffic speed statistics on the highways of Los Angeles county and the Bay Area respectively. We also evaluate in the publicly available Solar-Energy, Traffic, Electricity and Exchange-Rate. Specifications for each dataset are presented in Table 2, where #Nodes is the number of time series

|  | #Nodes | # Samples | Context length | Pred. length |
|---|---|---|---|---|
| METR-LA | 207 | 34,272 | 12 | 12 |
| PEMS-BAY | 325 | 52,116 | 12 | 12 |
| Solar-Energy | 137 | 52,560 | 168 | 1 |
| Traffic | 862 | 17,544 | 168 | 1 |
| Electricity | 321 | 26,304 | 168 | 1 |
| Exchange-Rate | 8 | 7,588 | 168 | 1 |

Table 2: Dataset specifications.

in the panel, #Samples is the number of time steps in each time series, Context length is the length of the input window and Pred. Length is the length of the predicted window. We compare to a variety of baselines including previous works and variations of our proposed method. We distinguish the following three main types of baselines (Univariate, Multivariate with known graph and Multivariate with graph inference):

- Univariate: In this case a single model is trained for all time series but they are treated independently without message exchange among them. These baselines include a simple linear Auto Regressive model (AR) and a variation of our FC-GNN method that we called NE-GNN where all edges have been removed such that the aggregation function AGG becomes equivalent to a multilayer perceptron defined by $\phi_h$ in Equation 3.

- Multivariate with a known graph: These methods require a previously defined graph, therefore they are restricted to those datasets where an adjacency matrix can be pre-defined (e.g. METR-LA, PEMS-BAY). From this group we compare to DCRNN (Li et al., 2017), STGCN (Yu et al., 2017) and MRA-BGCN (Chen et al., 2020a).

- Multivariate with graph inference or attention: These methods exchange information among different time series by simultaneously inferring relations among them or by attention mechanisms. From this group we compare to LDS (Franceschi et al., 2019), LST-Skip (Lai et al., 2018), TPA-LSTM (Shih et al., 2019), MTGNN (Wu et al., 2020) and GTS (Shang et al., 2021). Graph WaveNet (Wu et al., 2019) also belongs to this group but unlike the others it jointly uses a pre-defined adjacency matrix. Comparisons to NRI (Kipf et al., 2018) can be found in previous literature (Shang et al., 2021; Zügner et al., 2021). We additionally include a variation of our FC-GNN without unique node ids, we denote it by *(w/o id)* beside the model name. GTS numbers have been obtained by averaging over 3 runs its official implementation. https://github.com/chaoshangcs/GTS.

### 5.2 MAIN RESULTS

In this section we evaluate our method in METR-LA and PEMS-BAY datasets. For this experiment we used the training setup from GTS (Shang et al., 2021) which uses the dataset partitions and evaluation metrics originally proposed in (Li et al., 2017). The model has been trained by minimizing the Mean Absolute Error (MAE) between the predicted and the ground truth samples. The reported

| | METR-LA | | | | | | | | | PEMS-BAY | | | | | | | | |
| | 15 min | | | 30 min | | | 60 min | | | 15 min | | | 30 min | | | 60 min | | |
| | MAE | RMSE | MAPE | MAE | RMSE | MAPE | MAE | RMSE | MAPE | MAE | RMSE | MAPE | MAE | RMSE | MAPE | MAE | RMSE | MAPE |
|---|---|---|---|---|---|---|---|---|---|---|---|---|---|---|---|---|---|---|
| Linear / AR | 3.81 | 8.80 | 9.13% | 4.94 | 11.14 | 12.17% | 6.30 | 12.91 | 16.72% | 1.59 | 3.41 | 3.27% | 2.15 | 4.87 | 4.77 % | 2.97 | 6.65 | 7.03% |
| DCRNN | 2.77 | 5.38 | 7.30% | 3.15 | 6.45 | 8.80% | 3.60 | 7.60 | 10.50% | 1.38 | 2.95 | 2.90% | 1.74 | 3.97 | 3.90% | 2.07 | 4.74 | 4.90% |
| STGCN | 2.88 | 5.74 | 7.6% | 3.47 | 7.24 | 9.6% | 4.59 | 9.40 | 12.7% | 1.36 | 2.96 | 2.9% | 1.81 | 4.27 | 4.2% | 2.49 | 5.69 | 5.8% |
| MRA-BGCN | 2.67 | **5.12** | **6.8%** | 3.06 | 6.17 | 8.3% | 3.49 | 7.30 | 10.0% | **1.29** | **2.72** | 2.9% | **1.61** | **3.67** | 3.8% | **1.91** | **4.46** | 4.6% |
| Graph WaveNet* | 2.69 | 5.15 | 6.90% | 3.07 | 6.22 | 8.37% | 3.53 | 7.37 | 10.01% | 1.30 | 2.74 | **2.73%** | 1.63 | 3.70 | **3.67%** | 1.95 | 4.52 | 4.63% |
| LDS | 2.75 | 5.35 | 7.1% | 3.14 | 6.45 | 8.6% | 3.63 | 7.67 | 10.34% | 1.33 | 2.81 | 2.8% | 1.67 | 3.80 | 3.8% | 1.99 | 4.59 | 4.8 % |
| MTGNN | 2.69 | 5.18 | 6.86% | 3.05 | 6.17 | 8.19% | 3.49 | 7.23 | 9.87% | 1.32 | 2.79 | 2.77% | 1.65 | 3.74 | 3.69% | 1.94 | 4.49 | **4.53%** |
| GTS | 2.64 | 5.19 | 6.79% | 3.06 | 6.30 | 8.24% | 3.56 | 7.55 | 9.95% | 1.35 | 2.84 | 2.85% | 1.67 | 3.82 | 3.80% | 1.96 | 4.53 | 4.62% |
| *Ablation study* | | | | | | | | | | | | | | | | | | |
| NE-GNN *(w/o id)* | 2.80 | 5.73 | 7.50% | 3.40 | 7.15 | 9.74% | 4.22 | 8.79 | 13.06% | 1.40 | 3.03 | 2.92% | 1.85 | 4.25 | 4.21% | 2.39 | 5.50 | 5.93% |
| FC-GNN *(w/o id)* | 2.77 | 5.65 | 7.39% | 3.36 | 7.02 | 9.59% | 4.14 | 8.64 | 12.70% | 1.39 | 3.00 | 2.88% | 1.82 | 4.18 | 4.11% | 2.32 | 5.35 | 5.71% |
| NE-GNN | 2.69 | 5.57 | 7.21% | 3.14 | 6.74 | 9.01% | 3.62 | 7.88 | 10.94% | 1.36 | 2.88 | 2.86% | 1.72 | 3.91 | 3.93 % | 2.07 | 4.79 | 5.04% |
| *Our models* | | | | | | | | | | | | | | | | | | |
| FC-GNN | **2.60** | 5.19 | **6.78%** | **2.95** | **6.15** | **8.14%** | **3.35** | **7.14** | **9.73%** | 1.33 | 2.82 | 2.79% | 1.65 | 3.75 | **3.72%** | 1.93 | **4.46** | **4.53%** |
| BP-GNN (K=4) | 2.64 | 5.37 | 7.07% | 3.02 | 6.42 | 8.46% | 3.40 | 7.32 | 9.91% | 1.33 | 2.82 | 2.80% | 1.66 | 3.78 | 3.75% | 1.94 | **4.46** | 4.57% |

Table 3: Benchmark on METR-LA and PEMS-BAY datasets. Mean Absolute Error (MAE), Mean Squared Error (RMSE) and Mean Absolute Percentage Error (MAPE) are reported for different time horizons {15, 30, 60} minutes. Results have been averaged over 5 runs.

metrics are MAE, Root Mean Squared Error (RMSE) and Mean Absolute Percentage Error (MAPE) from (Li et al., 2017). All metrics have been averaged over 5 runs. All our models (FC-GNN, BP-GNN and NE-GNN) contain 2 graph convolutional layers, 64 features in the hidden layers, Swish activation functions (Ramachandran et al., 2017) and have been trained with batch size 16. The number of auxiliary nodes for BP-GNN was set to $K = 4$. Time experiments report the average forward pass in seconds for a batch size 16 in a Tesla V100-SXM GPU. Further implementation details are provided in Appendix B.1.

**Results** are reported in tables 11 and 4, results with standard deviations are reported in Appendix B.3. FC-GNN outperforms other methods in most metrics while being computationally cheaper than previous works. On the other hand, BP-GNN performs very competitively w.r.t. previous works (even outperforming all previous methods in some metrics) but with a vast decrease in computation. Furthermore, these performances are achieved without providing the structure information of the city (i.e. pre-defined adjacency) unlike in those methods that require it or in Graph Wavenet that optionally uses it. Additionally, notice the performance gap between FC-GNN and NE-GNN is larger when including a unique identifier of the nodes. This means the network can better leverage the information exchange among nodes when they are uniquely identified, but it still benefits from the message passing scheme when they are not uniquely identified (w/o id) thanks to the dynamical inference. Time results are presented in Table 4. BP-GNN is the most efficient algorithm in both METR-LA and PEMS-BAY by a large margin. FC-GNN is also more efficient than previous methods in both datasets but it is still limited by the $O(N^2)$ scalability. In Section 5.3 we will evaluate in a larger dataset showing that BP-GNN is even more efficient w.r.t. to other methods as the graph becomes larger thanks to its linear scalability $O(N)$. In this timing benchmark (Table 4) we included all methods from the previous Table 11 that have a publicly available implementation in METR-LA and PEMS-BAY datasets.

| | Forward Time (s) | |
| | METR-LA | PEMS-BAY |
|---|---|---|
| Linear | .0002 | .0002 |
| DCRNN | .2559 | .2754 |
| Graph WaveNet | .0500 | .0673 |
| MTGNN | .0160 | .0371 |
| GTS | .0869 | .1087 |
| NE-GNN | .0033 | .0047 |
| FC-GNN | .0108 | .0253 |
| BP-GNN (K=4) | .0044 | .0046 |

Table 4: Forward time in seconds for different methods.

## 5.3 SINGLE STEP FORECASTING

In this section, we evaluate our method in the publicly avilable Solar-Energy, Traffic, Electricity and Exchange-Rate datasets. In contrast to METR-LA and PEMS-BAY, these datasets do not contain spatial information from which a graph can be pre-defined, therefore, methods that rely on a known graph are not directly applicable. We use the same training settings as (Lai et al., 2018; Shih et al.,

| Dataset | | | Solar-Energy | | | | Traffic | | | | Electricity | | | | Exchange-Rate | | | |
|---|---|---|---|---|---|---|---|---|---|---|---|---|---|---|---|---|---|---|
| | | | Horizon | | | | Horizon | | | | Horizon | | | | Horizon | | | |
| Methods | Metrics | #Top2 | 3 | 6 | 12 | 24 | 3 | 6 | 12 | 24 | 3 | 6 | 12 | 24 | 3 | 6 | 12 | 24 |
| AR | RSE | (0) | .2435 | .3790 | .5911 | .8699 | .5991 | .6218 | .6252 | .6293 | .0995 | .1035 | .1050 | .1054 | .0228 | .0279 | .0353 | .0445 |
| | CORR | (0) | .9710 | .9263 | .8107 | .5314 | .7752 | .7568 | .7544 | .7519 | .8845 | .8632 | .8591 | .8595 | .9734 | .9656 | .9526 | .9357 |
| RNN-GRU | RSE | (0) | .1843 | .2559 | .3254 | .4643 | .4777 | .4893 | .4950 | .4973 | .0864 | .0931 | .1007 | .1007 | .0226 | .0280 | .0356 | .0449 |
| | CORR | (0) | .9843 | .9690 | .9467 | .8870 | .8721 | .8690 | .8614 | .8588 | .9283 | .9135 | .9077 | .9119 | .9735 | .9658 | .9511 | .9354 |
| LST-skip | RSE | (0) | .1843 | .2559 | .3254 | .4643 | .4777 | .4893 | .4950 | .4973 | .0864 | .0931 | .1007 | .1007 | .0226 | .0280 | .0356 | .0449 |
| | CORR | (0) | .9843 | .9690 | .9467 | .8870 | .8721 | .8690 | .8614 | .8588 | .9283 | .9135 | .9077 | .9119 | .9735 | .9658 | .9511 | .9354 |
| TPA-LSTM | RSE | (3) | .1803 | .2347 | .3234 | .4389 | .4487 | .4658 | .4641 | .4765 | .0823 | .0916 | .0964 | .1006 | **.0174** | **.0241** | .0341 | **.0444** |
| | CORR | (4) | .9850 | .9742 | .9487 | .9081 | .8812 | .8717 | .8717 | .8629 | .9439 | .9337 | .9250 | .9133 | **.9790** | **.9709** | **.9564** | **.9381** |
| MTGNN | RSE | (5) | .1778 | .2348 | .3109 | .4270 | .4162 | .4754 | **.4461** | **.4535** | .0745 | **.0878** | **.0916** | **.0953** | .0194 | .0259 | .0349 | .0456 |
| | CORR | (6) | .9852 | .9726 | .9509 | .9031 | .8963 | .8667 | **.8794** | **.8810** | .9474 | .9316 | .9278 | .9234 | **.9786** | **.9708** | **.9551** | **.9372** |
| NE-GNN | RSE | (2) | .1898 | .2580 | .3472 | .4441 | .4212 | .4586 | .4679 | .4743 | .0762 | .0917 | .0966 | .0994 | .0175 | **.0244** | **.0338** | .0447 |
| | CORR | (0) | .9829 | .9663 | .9367 | .8905 | .8951 | .8748 | .8700 | .8670 | .9494 | .9362 | .9308 | .9262 | .9769 | .9686 | .9535 | .9352 |
| FC-GNN | RSE | **(12)** | **.1651** | **.2202** | **.2981** | **.3997** | **.4057** | **.4395** | **.4624** | **.4620** | **.0732** | .0907 | **.0915** | **.0979** | **.0174** | .0245 | .0344 | .0450 |
| | CORR | **(12)** | **.9876** | **.9765** | **.9551** | **.9148** | **.9024** | **.8850** | **.8764** | **.8751** | **.9521** | **.9404** | **.9351** | **.9294** | .9772 | .9685 | .9538 | .9349 |
| BP-GNN (K=4) | RSE | (11) | .1704 | .2257 | .3072 | .4050 | .4095 | .4470 | .4640 | .4641 | .0740 | .0898 | .0940 | .0980 | .0175 | **.0244** | .0339 | **.0442** |
| | CORR | (10) | .9865 | .9751 | .9522 | .9138 | .8999 | .8820 | .8744 | .8723 | .9519 | .9396 | .9345 | .9288 | .9769 | .9684 | .9530 | .9360 |

Table 5: Benchmark on Solar-Energy, Traffic, Electricity and Exchange-Rate. Root Relative Squared Error (RSE) and Empirical Correlation Coefficient (CORR) are reported for different horizons {3, 6, 12, 24}. All results have been averaged over 5 runs. #Top2 column counts how many metrics in each row are in the top 2 (i.e. bold).

2019; Wu et al., 2020) firstly proposed by TLST-skip (Lai et al., 2018). In this experiment the network is trained to predict only one time step into the future (single step forecasting) with a given horizon (3, 6, 12, or 24) by minimizing the Mean Absolute Error (MAE). All datasets have been split in 60%/20%/20% for training/val/test respectively. As proposed in (Lai et al., 2018) we use the Root Relative Squared Error (RSE) and Empirical Correlation Coefficient (CORR) as evaluation metrics (both defined in Appendix C.2). All our models (FC-GNN, BP-GNN and NE-GNN) contain 2 graph convolutional layers and 128 features in the hidden layers. The timing results have been run with batch size 16. Further implementation details are provided in Appendix C.1. Results with standard deviations are provided in Appendix C.3

**Results** in Table 5 are consistent with the previous experiment. FC-GNN outperforms all previous works in most metrics. There is a significant improvement w.r.t. its cousin baseline NE-GNN, this demonstrates that sharing information among nodes is beneficial (except in Exchange-Rate dataset which only contains 8 nodes). BP-GNN performs better than previous methods in most metrics and very close to FC-GNN. Regarding the timing results (Table 6), BP-GNN is the most efficient graph inference method by a large margin in most datasets. The larger the number of nodes in the dataset, the larger the computational improvement of BP-GNN w.r.t. other methods due to its linear scalability. For example, in Solar-Energy (137 nodes), BP-GNN is 1.43 times faster than FC-GNN and 1.92 times faster than MTGNN. On the other hand, in a larger dataset as Electricity (321 nodes) BP-GNN is 6.38 times faster than FC-GNN and 9.18 times faster than MTGNN. In the largest dataset, Traffic (862 nodes), BP-GNN becomes 34.58 times faster than FC-GNN and 97.59 times faster than MTGNN, although in traffic, MTGNN and FC-GNN did not fit in the GPU for a batch of 16 (also because to the $O(N^2)$ complexity), and we had to pass the samples in batches of 2 which eliminates part of the GPU parallelization. Running BP-GNN in batches of 2, would result in it being 7.71 and 18.92 times faster than FC-GNN and MTGNN respectively. Finally, in such small graphs as Exchange Rate, there is no computational benefit in using the bipartite assumption since the number of edges for both the bipartite and the fully connected graphs becomes the same (for K=4 and N=8).

| | Exchange | Solar | Electricity | Traffic |
|---|---|---|---|---|
| # Nodes | 8 | 137 | 321 | 862 |
| MTGNN | .0062 | .0146 | .0771 | 1.1808 |
| NE-GNN | .0034 | .0059 | .0067 | .0117 |
| FC-GNN | .0053 | .0109 | .0536 | .4184 |
| BP-GNN | .0076 | .0076 | .0084 | .0121 |

Table 6: Forward time in seconds for different methods in Solar-Energy, Traffic, Electricity and Exchange-Rate datasets.

## 5.4 INFERRED GRAPH ANALYSIS

In this section we study the adjacency matrices inferred by our FC-GNN and BP-GNN methods. For this purpose, we created two synthetic datasets of $N = 10$ time series and $T = 10.000$ timesteps each.

(a) "Cycle Graph Gaussians" samples each series value $\mathbf{x}_{i,t}$ from the past $(t-5)$ of another series $(i-1 \mod N)$ from the panel. The resulting adjacency matrix is a directed cycle graph. More formally, the dataset is generated from the following gaussian distribution $\mathbf{x}_{i,t} \sim \mathcal{N}(\beta\mathbf{x}_{i-1 \mod N,t-5}; \sigma^2)$, where $\beta = 0.9$ and $\sigma = 0.5^2$. (b) We call the second dataset "Noisy Sinusoids", inspired by the Discrete Sine Transformation we generate arbitrary signals as the sum of different sinusoids plus gaussian noise. We share the same sinusoids among the first half $1 \leq i \leq 5$ of the panel and among the second half $6 \leq i \leq 10$ while sampling different gaussian noise for each signal $i$. This creates strong dependencies among the signals within each half. The expected adjacencies are plotted in Figure 2. Further details and visualizations of these two datasets are provided in Appendix D.1. Both BP-GNN and FC-GNN consist of a single graph convolution from which adjacencies are obtained. We predict the next time step into the future and optimize the MAE during training. Since our graph inference mechanism is dynamic we average them over 10 timesteps $t$. For BP-GNN we construct the adjacencies as the first N rows and columns of $\tilde{A} = A_2 A_1$ (Eq. 7, Sec. 4). Notice that small K values in BP-GNN may enforce sharing auxiliary nodes and in consequence a more dense adjacency $\tilde{A}$. Further implementation details are described in Appendix D.2.

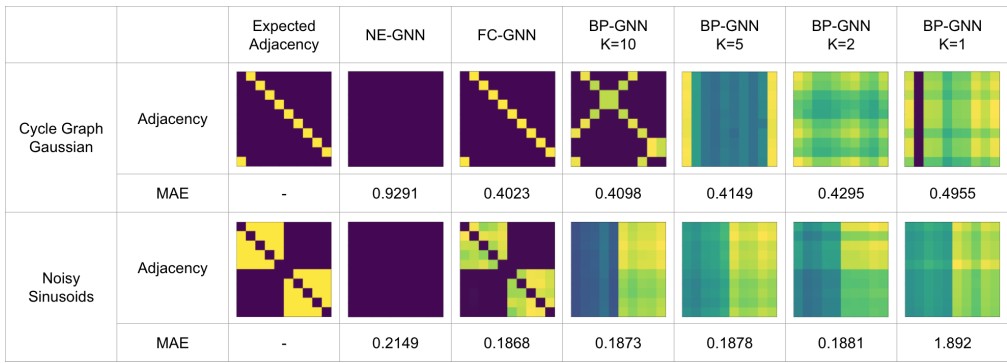

| | | Expected Adjacency | NE-GNN | FC-GNN | BP-GNN K=10 | BP-GNN K=5 | BP-GNN K=2 | BP-GNN K=1 |
|---|---|---|---|---|---|---|---|---|
| Cycle Graph Gaussian | Adjacency | | | | | | | |
| | MAE | - | 0.9291 | 0.4023 | 0.4098 | 0.4149 | 0.4295 | 0.4955 |
| Noisy Sinusoids | Adjacency | | | | | | | |
| | MAE | - | 0.2149 | 0.1868 | 0.1873 | 0.1878 | 0.1881 | 1.892 |

Figure 2: Inferred adjacency matrices and MAE losses in the proposed synthetic datasets.

**Results** are reported in Figure 2. FC-GNN perfectly matches the ground truth adjacencies and it has the lowest MAE test loss. In the "Cycle Graph Gaussians" dataset, BP-GNN infers a similar matrix to the ground truth when provided with enough auxiliary nodes $K = 10$, but the matrix becomes denser when reducing $K$, which is expected since different messages are forced to share the same auxiliary nodes. Despite that, in both datasets, BP-GNN manages to obtain very competitive performance (even for small $K$) by inferring suboptimal matrices with denser connections. In the "Noisy Sinusoids" dataset, the benefit of increasing $K$ is smaller, this is coherent with the fact the ground truth matrix is denser. In all cases the proposed models significantly outperform their non-edge variant NE-GNN. Further visualizations for different random seeds are provided in Appendix D.3.

## 5.5 CHOOSING THE NUMBER OF AUXILIARY NODES K

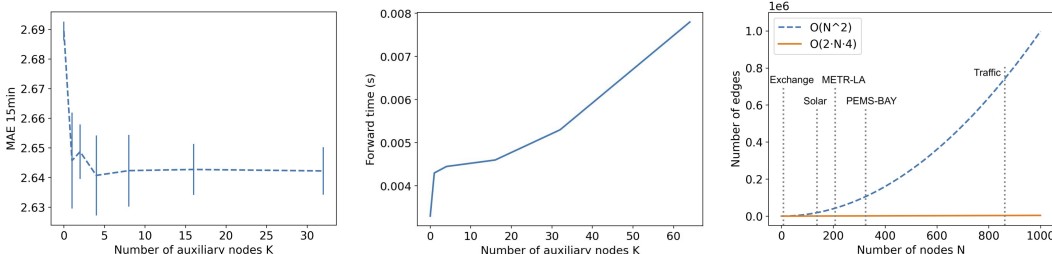

Figure 3: (Left) Mean Absolute Error in METR-LA for different K values (10 runs average). (Middle) Running times in METR-LA. (Right) Number of edges w.r.t to the number of nodes.

In the previous section we evaluated BP-GNN for different $K$ values, and we found small values were enough to achieve good performance in the denser adjacency dataset (Noisy Sinusoids). In real world scenarios, we found a similar behavior, where results are competitive with small $K$ values. Therefore, we chose a relatively small number of auxiliary nodes $K = 4$ that resulted in a good

trade-off accuracy vs computation. In Figure 3, we plot the MAE for different $K$ values in METR-LA for a 15 min time horizon, and we see the best performances are already achieved at a small $K$ values. In the middle plot of Figure 3, we report the forward running time in (s) for a batch size 16 for METR-LA as we sweep different K values. Finally, the right plot of Figure 3 shows the scalability of the number of edges w.r.t. to the number of nodes for $O(N^2)$ (e.g. FC-GNN) and $O(2N4)$ (i.e. BP-GNN (K=4)). This illustrates the vast difference in scalability as the number of nodes increases.

## 6 Conclusions

We presented a novel approach for multi-variate time series forecasting which lends itself to easy integration into existing univariate approaches by ways of a graph neural network (GNN) component. This GNN infers a latent graph in the space of the embeddings of the univariate time series and can be inserted in many neural-network based forecasting models. We show that this additional graph/dependency structure improves forecasting accuracy both in general and in particular when compared to the state of the art. We alleviate typical scalability concerns for GNNs via allowing the introduction of auxiliary nodes in the latent graph which we construct as a bipartite graph that bounds the computational complexity. We show that this leads to computationally favorable properties as expected while the trade-off with forecasting accuracy is manageable.

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
