# OpenReview forum: "Multivariate Time Series Forecasting with Latent Graph Inference"
_ICLR.cc/2022/Conference — ICLR 2022 Submitted_

### Official Review · Reviewer_CU8M · 2021-11-01

**Correctness:** 2
**Technical Novelty And Significance:** 2
**Empirical Novelty And Significance:** 2
**Recommendation:** 3
**Confidence:** 5

**Main Review:**

The paper is fairly well-written and its motivation is clear. It is also technically correct. The authors tried their proposed method on a few small datasets.

The FC-GNN version proposed in the submitted paper is neither novel nor significant. The main argument of the paper is reducing the computational complexity of the relational inference, which I think is a valid motivation for this paper, but I would have rather expected to try a more challenging dataset with a large number of nodes for achieving the same goal, i.e. see if the reduction in computational complexity in this way is helpful in a practical manner or not. Inferring relations between a small number of features, i.e. 8 to 800, is not a challenging problem, and other papers such as FNP (the functional neural processes, NeurIPS 2019) and BayReL (Bayesian Relational Learning, NeurIPS 2020) inferred a much larger relational graph (structured latent space) with a simple method $O(N^2)$, however, it is really challenging to learn a graph with 20,000 nodes.

Even the BP-GNN model proposed in this paper is learning a Gaussian mixture in my opinion, as it will learn the connection weight of each node with Gaussian auxiliary nodes, leading to having a Gaussian mixture latent space. I believe the authors at least need to show that adding such a computational learning $O(NK)$ is improving the performance compared to a mixture of Gaussian latent space as a baseline.

Apart from that, the authors are learning one graph for a period of time. In my opinion, the inferred graph only captures high-level information, it might miss the short-term dependencies. An ablation study on this might be very helpful similar to those that have been done on NLP.

I also believe the authors still are able to compare their model with NRI. In addition, for those methods "Multivariate with a known graph", the authors can construct ad-hoc graphs such as identity and/or fully connected graphs to compare the performance.

I don’t find the plots in Figure 2 very insightful and it is quite hard to see any informative insights regarding BP-GNN in my opinion. I encourage the authors to create an ablation study based on the BP-GNN and see if they can reconstruct the bipartite graph. Even for the fully connected graph, I prefer to see a real example that you have the graph and see if the method can learn the graph. The authors should be able to find some data in traffic applications for example.

The computational complexity needs to be discussed in the main paper as it is the main argument of the paper.

Comparisons with GRNN and RNN methods are needed in which you have the true graph or you do not have it to see if learning the graph dependency is helpful or not.

The standard deviation of the performance should be reported, considering the results are very close to each other.

**Summary Of The Paper:**

The paper proposes to infer a relational latent space in order to do multi-variate time series forecasting. The main contribution is reducing the computational complexity of the inferred graph by converting the fully connected graph learning problem to a bipartite graph learning method in which the nodes are connected to K auxiliary nodes.

**Summary Of The Review:**

Overall, I think that this paper would need to be extremely solid on the experimental side with potentially further experiments, e.g. using a larger number of features, robustness to missing data, etc, in order to make up for the somewhat limited contributions in terms of novelty.

---

> ### Author Response · Authors · 2021-11-17
> **Response B to Reviewer CU8M**
>
> >I also believe the authors still are able to compare their model with NRI.
>
> Previous works [1, 2] already compared to NRI in the context of multivariate time series forecasting. They show the baselines GTS and MTGNN (included in our work) outperform NRI. Additionally, they were only able to compare in the small datasets of our paper, reporting NRI was not able to scale to the larger ones. We do however mention NRI as an important method in our discussion of related work. Taking reviewer’s feedback, we have mentioned these comparisons from previous works in the updated manuscript.
> [1] Shang, C., Chen, J., & Bi, J. (2021). Discrete Graph Structure Learning for Forecasting Multiple Time Series.
> [2] Zügner, Daniel, et al. (2021). A Study of Joint Graph Inference and Forecasting.
>
> >In addition, for those methods "Multivariate with a known graph", the authors can construct ad-hoc graphs such as identity and/or fully connected graphs to compare the performance.
>
> A baseline such as identity graph was already included in the experiments section. This baseline was defined in Section 5.1 with the name NE-GNN. Additionally, based on the reviewer’s feedback, we will include a fully connected graph baseline in the final version of the paper.
>
> >I don’t find the plots in Figure 2 very insightful and it is quite hard to see any informative insights regarding BP-GNN in my opinion. I encourage the authors to create an ablation study based on the BP-GNN and see if they can reconstruct the bipartite graph. Even for the fully connected graph, I prefer to see a real example that you have the graph and see if the method can learn the graph.
>
> In the experiment of Figure 2 we report the inferred matrices in a synthetic dataset in which we have access to the ground truth adjacency matrix from which the data was generated. This makes the evaluation of the adjacency matrix less subjective than in real world datasets since here we can compare to the ground truth graph.
>
> Conversely, in a real world dataset as METR-LA, we have no access to the ground truth graph, even if we can “assume” a graph, for example, from the relative distances among the sensors in the city. Despite that, we agree with the reviewer that it would be interesting to analyze the inferred graphs in a real world dataset and we added a small experiment section in the Appendix analyzing the inferred graph for METR-LA.
>
> Additionally, after reviewer’s feedback, we also included the plots of a bipartite graph ($A_1$, $A_2$ from equation 7) instead of just the product $\hat{A}=A_2·A_1$ (eq. 7) in the Appendix.
>
> >The standard deviation of the performance should be reported
>
> Another reviewer had mentioned this also and after having read your feedback, we included all standard deviations in the Appendix section.

---

> ### Author Response · Authors · 2021-11-17
> **Response A to Reviewer CU8M**
>
> Thank you for the feedback and for indicating that the paper is well-written and technically correct. We answer the raised issues below.
>
> >Correctness of the paper
>
> While the reviewer states the paper is technically correct, the correctness score in the review is the lowest (⅕).  We find this contradictory and we wonder if this may be a mistake.
>
> >Reviewer states that while one of the main arguments is reducing the computational complexity, the dataset with the largest number of nodes in the experiment section contains only 862 nodes. And it would be more interesting to scale to larger graphs, e.g. 20,000 nodes.
>
> We agree with the reviewer that it would be very interesting to analyze the performance in graphs of 20,000 nodes. We are currently not aware of a public time series data set of this size and would appreciate a pointer, in particular if other related work compared on it. We speculate that most of the methods we compare with would be difficult to scale the training to data sets of this size due to their quadratic complexity.
> We think some of the datasets used in our evaluations are already large enough to show the computational benefits of our method. For example, in Traffic dataset (862 nodes), our FC-GNN method is 2.82 times faster than previous work MTGNN and our BP-GNN is 18.92 times faster than MTGNN. This efficiency is consistent in smaller datasets too, for example, FC-GNN and BP-GNN are 1.47 and 8.07 times faster than MTGNN in PEMS-BAY (325 nodes), while MTGNN is the fastest graph inference work in the table.
>
>
> >The computational complexity needs to be discussed in the main paper as it is the main argument of the paper.
>
> We are not completely sure about this question and kindly ask for clarification. We want to point out that in our current manuscript, the complexities of the proposed method $O(N^2)$, $O(NK)$ have been explained. If this was buried in the exposition, we apologize and we could make this more clear. Additionally, as mentioned above, different timing benchmarks  (tables 4 and 6) are reported in the main experiments showing our methods outperform previous methods in terms of running time. If this doesn’t answer the original concern of the reviewer, we are happy to provide more details.
>
> >The BP-GNN model proposed in this paper is learning a Gaussian mixture in my opinion, as it will learn the connection weight of each node with Gaussian auxiliary nodes, leading to having a Gaussian mixture latent space.
>
> We kindly ask for clarification about this question. Our BP-GNN is defined as a Graph Neural Network that exchanges non-linear messages on a bipartite graph where K auxiliary nodes “route” and store  information among the N nodes in the time series. There are no Gaussian constraints in the embedding of these nodes, and even if they were, it is not clear to us how the method would be similar to a Gaussian mixture. We would be happy to clarify this point in the related work once we understand the reviewer’s intention a bit more.
>
> >Apart from that, the authors are learning one graph for a period of time. In my opinion, the inferred graph only captures high-level information, it might miss the short-term dependencies.
>
> The inferred graph is computed from the input windowed data $x_{t_0:t}$ (short-term) and from the unique id of the time series which doesn’t depend on time $t$ (long-term). Therefore, the network can potentially leverage short and long term dependencies depending on what is more beneficial when optimizing on the given task/dataset. For example, in the provided synthetic dataset (Section 5.4), dependencies do not change over time, therefore the inferred graph is very similar regardless of the windowed data $t$. On the other hand the baseline “FC-GNN (w/o id)” in Section 5.2, can only leverage short term dependencies since unique ids are not included, and there is an improvement in performance w.r.t no using relational information.
>
> We think this conclusion from the reviewer may be drawn from the plots of experiment 5.4 where the relations are the same for every $t$. In order to show it doesn’t have to be the case in every dataset, we have included in the appendix the inferred relations $\alpha_{ij}$ for METR-LA for different time steps $t$, showing that even if the overall graph has some similarities over $t$, some components differ for different $t$.

---

> > ### Comment · Reviewer_CU8M · 2021-11-28
> > **RE: Response A to Reviewer CU8M**
> >
> > Thanks for the response.
> >
> > **Correctness of the paper** While I have updated my score, I still believe the claims are not well supported, neither theoretically nor empirically.
> >
> > **We are currently not aware of a public time-series data set of this size and would appreciate a pointer** I still believe this should be included through synthetic if the real dataset is not available.
> >
> > **Discussing computational complexity** The authors are only compared the computational complexities between the two proposed models. However, a comparison with the other existing models is needed.  If the authors plan to report the run time, it should be training time and even that might not be completely fair if they are not implemented in the same way. However, the authors only provide the forward run time which for me is not a good way to compare the computational complexity.
> >
> > **Gaussian mixture** I agree with the authors' response. I have read the paper one more time as well as the authors' response and found my previous understanding was incorrect. I would suggest the authors re-write this paragraph as it is not clear.
> >
> > I still believe the paper has a lack of novelty. Overall, I think that this paper would need to be extremely solid on the experimental side with potentially further experiments in order to make up for the somewhat limited contributions in terms of novelty. For now, I cannot recommend this paper for acceptance at ICLR.

---

### Official Review · Reviewer_kzU4 · 2021-11-01

**Correctness:** 3
**Technical Novelty And Significance:** 3
**Empirical Novelty And Significance:** 3
**Recommendation:** 5
**Confidence:** 3

**Main Review:**

Among the strengths we can think of the method, and the representation. Namely:
Methods that account for relations due to confounders or other relations are important to consider and directly incorporating them in time series analysis  could avoid using regularization during model fitting.
The latent graph inference idea could be useful to automatically identify the relations existent in a multi-variate time series model more broadly; for instance, in partially observed settings.
With respect to the weaknesses, while the paper is relatively well organized, there are many details that are not clear. In particular, the architecture is presented and its components listed, but the connection of these components and the overall goal of the paper is not discussed in enough detail as to justify the choices. For instance how are colinearities identified by the architecture? How are non-linear dependencies discovered and summarized by the architecture? What other confounding can the architecture detect?

**Summary Of The Paper:**

This paper presents a method to combine information of multivariate time series by extending univariate architectures. The technique uses a graph representation to represent interactions by assuming a bipartite structure, which allows the technique to scale the representation to reduce the complexity from $O(N^2)$ to $O(N K)$ using K additional nodes. The architecture represents each time series as one of N nodes and associate K embeddings (nodes) and thus there are connections between the N nodes and the K nodes but not within each group of nodes. The architecture encodes the confounders, co-linearities, and other information among the time series to improve forecasting. The experiments show the performance (and time efficiency) of the technique against several baselines on METR-LA and PEMS-BAY datasets. The experiments also show the performance on single-step forecasting on four publicly available datasets. Finally, synthetic datasets were used to evaluate the adjacency matrices in control scenarios.


**Summary Of The Review:**

The paper has some strengths that could be useful in applications of multivariate time series forecasting. The graph representation could be useful not only for prediction but also for analysis and interpretability of results. However, I feel that more can be said about the architecture and how it affects the type of relations that it is capable of represent.

---

> ### Author Response · Authors · 2021-11-17
> **Response to Reviewer kzU4**
>
> First of all, thank you for the feedback. We answer the raised issues below.
>
> >The architecture is presented and its components listed, but the connection of these components and the overall goal of the paper is not discussed in enough detail as to justify the choices.  For instance how are colinearities identified by the architecture? How are non-linear dependencies discovered and summarized by the architecture? However, I feel that more can be said about the architecture and how it affects the type of relations that it is capable of representing.
>
> Our goal in the present work is to have a simple, efficient, modular forecasting model with competitive performance. Our proposal is modular in the sense that we can combine it with most sequence-to-sequence forecasting models, it is efficient because of the bi-partite graph construction and because messages are only exchanged at time step “t”, and it is competitive in terms of accuracy as shown throughout the experiments.
>
> To answer your question in more detail, the model can potentially leverage linear and non-linear dependencies among nodes. This is achieved by using a Graph Neural Network that exchanges messages $m_{ij}$ among pairs of nodes. Those messages $m_{ij}$ are the outptut of a non-linear function approximated by an MLP that takes as input the embedding of pairs of nodes (i,j) and outputs the mentioned non-linear message $m_{ij}$. Additionally, messages $m_{ij}$ are “gated” or multiplied by a scalar value in the range $\alpha_{ij} \in (0, 1)$. This gating mechanism $\alpha_{ij}$ is also inferred by the network.
>
> Therefore, linear and non-linear dependencies can be modelled by $m_{ij}$ and $\alpha_{ij}$. In terms of explainability, $m_{ij}$ is an non-linear embedding that can embed non-linear information for each edge, on the other hand $\alpha_{ij}$ is a single scalar that can be more interpretable, for example, $\alpha_{ij} = 0$ implies the network decided to gate out that connection. In experiment 5.4, we can see that in the FC-GNN scenario and in some BP-GNN scenarios, $\alpha_{ij}$ is set to 0 when there aren’t dependencies among nodes, which means messages are gated out.
>
> Based on your feedback, we have extended the Method Section explaining in more detail how the proposed architecture choices are useful to model the non-linear co-dependencies of the data. Additionally, we have also included a section in the Appendix to show how the relations $\alpha_{ij}$ among nodes are inferred in a real world dataset as METR-LA.

---

### Official Review · Reviewer_xjBF · 2021-11-02

**Correctness:** 3
**Technical Novelty And Significance:** 3
**Empirical Novelty And Significance:** Not applicable
**Recommendation:** 5
**Confidence:** 4

**Main Review:**

**Strengths:**
1. The approach is interesting and has some novel elements, especially using a latent bipartite graph as a step towards better scalability.

Enabling multivariate forecasting with unknown relationships among series while also addressing the scalability challenge is an important problem in forecasting, and this is a promising direction.

Additionally I really liked how the authors posed the idea as a simple add-on to univariate methods, to be able to tack on multivariate modeling to a given univariate method - though it may have been interesting to actually test this idea out more in the experiments (as in, take some other or recent state of the art global univariate models and see if using this approach adding the graph neural net to it would help - for the given datasets and especially if there are some that really depend on multivariate models).


2. A large collection of datasets and experiments were performed, and an extensive set of graph neural net based forecasting methods compared with.


3. Additional useful ablation study, hyper parameter sensitivity study, and adjacency matrix result analyses was also provided.  This provides more interesting information and understanding of the proposed approaches.


**Weaknesses:**
1.  The complete method details and procedure is not clear from the description, and code is also not provided, so as it is the work does not seem fully reproducible.

a.) For the auxilliary nodes, U, it says they are initialized with random noise, but nothing else is said about them - i.e., how they are used in training and inference.  In particular, is this initialization done once at the beginning of training / initializing the entire model, and the values of U are fixed from then on to those values (for the rest of training and subsequent test prediction)?
Or are the embedding values updated as part of training as well?  Or are different random values used in each time window?  This is not clear.

b.) The description is a little confusing in a few places.  In particular, in Equation 3 and later references back to it - it would be helpful to provide the actual definitions of the various phi functions (at least in the appendix) and dimensions / example dimensions of  various inputs and outputs such as the embeddings.

c.) It would be helpful to understand the actual training process, even if in the appendix or said briefly - i.e., presumably sliding windows were used across the full input time series to generate each sample, presumably for every step size of 1?  Presumably 1 batch is just different time windows, containing all time series, so inputs are BxTxN where B is batch size, T is time window size, and N is the number of time series?

d.) Earlier when describing past work, context c is mentioned.  How is context c incorporated into the proposed model, as it seems absent from the method description?  What about exogenous series (i.e., context or features that vary with time but are not part of the series we are predicting)?


2. Experiment details are missing (e.g., based on the description I assume proper hyper parameter selection was not done) and results seem not too convincing, aside from speedup using the bipartite graph formulation.

a.) Various missing experiment details:
i. When describing the data, what is "Enc. Length" and "Dec. Length" in Table 2?  Is it "Enc. Length" the history window size used in each case, and "Dec. Length" the prediction window size?   If so that means for the first two datasets the next 12 values are being predicted, how then are different horizons evaluated / what is the meaning of the different horizons in the results table?  For the second set of datasets, why not evaluate predicting multiple time points (such as next 48 hours) as is commonly done in other past work, as opposed to single time point prediction?

ii. It would also be best to report the periodicity of the time series - i.e., is it daily, hourly, etc...

iii. Also details about the evaluation procedure are missing - in particular, was a sliding test window used to evaluate the performance, how many test windows were used, was re-training done before each, etc.?  Some of these may be able to be found in the pointed to prior work (saying it is the same setup) but really these details should be included in the appendix, and what was specifically done here for all particulars cannot be known from just the setup (such as if retraining is done or not).

iv. What context (c) or exogenous series were used in the experiments?

b.) As far as I can tell, since specific and different hyper parameters are reported in the appendix for each data set, it seems like the hyper parameters were used that gave the best results on the test set, which if so can be misleading and not reflect the actual performance.  Hyper parameters should be selected from a grid (or using HPO) based on validation data only.  I saw no mention of hyper parameter selection being performed in this manner, and I did not see reported the set of hyper parameters searched over for each method, so I have to assume it was not done.  If it was done, please report the set of hyper parameters searched over for each method, data set, and horizon.  Code would have also been useful to verify this here.

c.) Although 5 different runs were performed and averages across those were reported, no standard deviations are reported. The std. dev. should be reported as well, especially since in many cases there is only a tiny difference in scores compared to past methods.

d.) It is hard to read through the details of the results since there are so many, but from what I can tell it does not seem like the proposed method significantly improves over past methods - results seem practically the same or only tiny improvements.  It might help to present overall average results (for example mean and std. scores across all datasets as a final column, and plotting the different average metric scores with error bars vs. dataset size / number of series or horizon), percent improvements, and especially significance analyses (e.g., a critical difference diagram I think could be most useful here given the number of different methods and dataset+horizons - e.g., see https://mirkobunse.github.io/CriticalDifferenceDiagrams.jl/dev/ or https://github.com/hfawaz/cd-diagram).

e.) Comparisons to other classes of models would be nice to include as well - in particular good, recent global univariate and non-graph multivariate methods.
It would be nice to have representatives from the state of the art of global univariate methods (such as the recent transformer approaches such as Li et al. 2019 or more recent works that improved on this) in the comparison to see how they stack up, and validate the hypothesis that the multivariate modeling is even necessary in all these datasets.
Similarly it would be nice to see the comparison with recent non-graph neural net approaches, as they are just as relevant here, and there are recent approaches with state of the art results, some mentioned here, like, Salinas et al. 2019 and other more recent ones and ones focusing on scalable multivariate forecasting (mentioned below).  This would validate the hypothesis that using graphs is necessary / could help over these other approaches.


3. Novelty is somewhat limited.

As pointed out there are several works learning graph structure as part of forecasting (and some works on hierarchical structure, though not applied to forecasting - e.g., Ying, Rex, et al. "Hierarchical graph representation learning with differentiable pooling." NeurIPS 2018).  One additional missed work for learning graph structure as part of multivariate forecasting is the following:
* Cao, Defu, et al. "Spectral Temporal Graph Neural Network for Multivariate Time-series Forecasting." NeurIPS 2020.

Arguably the main novelty comes from introducing the latent bipartite graph in order to improve the scalability, however, the are other past works that also focus on improving scalability for multivariate forecasting, and that can learn the relationships scalably.
While I feel the approach proposed here is novel given the usage of graphs, I also feel this idea is in line with other prior work that also consider somewhat similar ways of enabling scalable multivariate forecasting (e.g., using a smaller set of latent node = series), and it may be worth mentioning this connection and approaches targeted at scalable multivariate forecasting.

One is mentioned - Salinas et al. 2019 - that similarly uses per-series model components to embed each series and combine them with a low-rank Gaussian copula, which enables more scalable modeling and training (even enabling sub-sampling among the series).
Another closely related line of work targeted at scalability is to use a smaller set of latent global series (similar in concept to the latent nodes introduced here since a node corresponds to a time series) - and modeling learns these latent global series (or how to derive them from the inputs) and the input series relationship to these latent global series to get predictions for individual time series.  These similarly enable improving the complexity of the forecast modeling part from O(N^2) to O(NK^2) using the K latent series.  E.g.:
* Sen et al. "Think globally, act locally: a deep neural network approach to high-dimensional time series forecasting." NeurIPS 2019.
* Nguyen et al. "Temporal Latent Auto-Encoder: A Method for Probabilistic Multivariate Time Series Forecasting." AAAI 2021.

It would be nice to compare and contrast against these as well, at least in text to call out the related line of work, and pros and cons.  Ideally it would also be nice to see a comparison in the experiments to some of these or else another recent state of the art multivariate forecasting method that is not considering graph structure / graph neural nets.


**Summary Of The Paper:**

The paper proposes using graph neural net (GNN) operations to combine per-series embeddings, to enable multivariate forecasting.

Specifically, N individual series are separately encoded for a given time window to get representations per series.  These representations are then updated with a GNN - either assuming a fully connected graph (with edge weights computed as part of the model), or using a bipartite graph (using a smaller set of K << N auxilliary nodes).  I.e., after multiple layers of the GNN, the representations are updated with information from the other series' representations.  Finally, the final representations are passed through per-series decoders.

The latent bipartite graph formulation enables more efficient operation of the GNN component as instead of computing O(N^2) messages only O(NK) need to be computed in a given pass.

The authors compare the proposed approach with other Graph based forecasting approaches (including ablated versions of the proposed approach) on 6 datasets - 2 where there is a given graph structure (so past work requiring the graph structure to be known can be used) and 4 where there is not.  They demonstrate competitive performance of the proposed approach (including the bipartite graph approach) and significant speed up of forward passes using the bipartite formulation especially for larger N.  They also examine the inferred adjacency matrices for different methods on some synthetic data.  Additionally, they show hyper parameter sensitivity results for varying K (number of latent nodes in the bipartite graph) for one dataset.

**Summary Of The Review:**

Overall I think the idea is interesting and in particular the direction of a latent bipartite graph as a graph neural net way of making multivariate forecasting more scalable is novel, however it could be useful to point out and contrast to related alternate approaches for scalable multivariate forecasting, which are similar but not using graph neural nets.

However there are several questions about the method and questions and concerns about the experiments that I feel need to be addressed before I can recommend the paper for acceptance - i.e., specifics of the method so it could be reproduced and understood by the readers, and details of the experiments to determine how hyper parameters were chosen and if the results show significant improvements.

Without additional clarifications and details around the experiments and the method I am leaning on the side of rejection.  However, if these can be addressed I could change my recommendation.



***Update after responses and discussions:***

I appreciate the responses, and read the author responses, updates, and other reviews and comments.  I tend to agree with the other reviewers' points and don't feel all concerns are addressed with the updates.

As pointed out the key novelty is really using the bipartite graph to increase computational efficiency, and as other reviewers have argued little analyses around this is done, larger data sets should be included, and training time scaling reported.  Despite the authors claim that larger data sets do not exist, and aside from simulated data, some larger ones exist and have been used in several other papers. E.g., the large wiki dataset has > 100,000 time series (and used in one of the scalable time series forecasting papers I cited and other past work it cites), and the recent M5 dataset and competition also has 30,490 inter-related time series at the lowest level, and there are several forecasting papers published about it as well.  Additionally more can be found with some searching / reading. The authors would have at least seen the wiki dataset if they looked at other prior papers including what I mentioned in my review.

In particular, as the main benefit they are claiming from their method is enabling scaling for multivariate forecasting to large amounts of time series, they ideally should consider related work on scaling forecasting for large collections of time series and at least discuss this area of work and contrast to their work, which they did not, despite the recommendation. They also should compare to the state-of-the-art multivariate forecasting methods (including without considering graphs, to demonstrate the graph approach is helpful), and especially need to compare to those designed to scale to many time series, which also was not done.

I also agree with the other reviewer - to only compare and report forward evaluation run times leaves uncertainty about practical performance, as training is the biggest issue in most cases.  Further, there seems little reason not to report it and explain what it reveals unless the benefits are not as much in this case - which would also be interesting and fine just needs to be reported and some analyses provided. Ideally the analyses around training and forward time should be performed for varying number of time series across multiple datasets, e.g., plotting on x-axis the number of time series, and on y-axis the training and forward time (and nice to also see the error metrics as well per number of time series).  It could even be combined in the same tables or plots for forward time by showing average training time per step and average forward time per step, for example, on dual y-axis if space is a concern) vs. number of time series.   This could potentially also be generated from larger datasets if needed by sub-sampling series, or just using the natural number of time series in different datasets.

The fact training time isn't reported makes one wonder if there could be some reasons it's not providing as much scaling benefits for training, e.g., maybe the train time is actually not much if any better, due to the extra propagation steps having more of an impact at training time as perhaps slowing down the back propagation updates (which we don't see from forward time alone), and also from the fact that all time series still must be present at once so there could be some dominating factor on the data side that doesn't change.

---

> ### Author Response · Authors · 2021-11-17
> **Response B to Reviewer xjBF**
>
>
> >2.a.iii. Was a sliding test window used to evaluate the performance, how many test windows were used, was re-training done before each, etc.?. Some of these may be able to be found in the pointed to prior work.
>
> We follow the approaches from prior works in detail here where those details are provided. However, we understand now after your comment that it would be better to make our work more self-contained, hence we have included those details in the Appendix.
>
> >2.b) As far as I can tell, since specific and different hyper parameters are reported in the appendix for each data set, it seems like the hyper parameters were used that gave the best results on the test set
>
> All hyperparameters have been tuned on the training and validation partitions. The test partition was never used to tune any parameters of the network, but only to report the final results. For example, different learning rates may be required for different datasets and models. We actually swept over the following subset of learning rates: lr $\in$ {1e-4, 2e-4, 5e-4, 1e-3, 2e-3, 5e-3} for each dataset and model. We had omitted some details on our hyperparameter tuning, but based on your feedback, we have included all hyperparameter searches in the appendix.
>
> >2.c) Although 5 different runs were performed and averages across those were reported, no standard deviations are reported. The std. dev. should be reported as well.
>
> We had omitted standard deviations for increased readability of the main results, but based on your feedback, we have included all standard deviations in the Appendix.
>
> >2.d) It is hard to read through the details of the results since there are so many. It might help to present overall average results
>
> We agree table results are quite big and a metric that summarizes all datasets to assess the global performance of each method would help interpreting them, specially for Table 5. We have included a summary of the table results in an extra column as proposed by the reviewer in Table 5.
>
> >2.e.) Comparisons to other classes of models would be nice to include as well - in particular good, recent global univariate and non-graph multivariate methods.
>
> We agree that Salinas 2019 would be an interesting global univariate baseline which we will consider. Regarding non-graph multivariate methods, we want to point out that some methods in the baseline are already multivariate and non-graph as (LST-Skip and TPA-LSTM).
>
> >3.) Novelty is somewhat limited. The main novelty comes from introducing the latent bipartite graph in order to improve the scalability.
>
> We understand the reviewers point of view and would argue that the perceived lack of novelty may in part be explained by the fact that our main contribution, the latent bipartite graph, is a simple idea (while being novel). We view this simplicity as important and consequential since it brings along benefits such as easy implementation and integration into existing methods.
> Besides the bipartite assumption, there are further contributions, where our method under the fully connected assumption outperforms previous models in most datasets while being much cheaper in computation time. And this is achieved with a (again) simple inference mechanism inside a standard GNN.
> We think this combination of good performance, cheap computation, modularity and simplicity can make the model very useful for the community even if its theoretical novelty and complexity is not the main selling point.
>
> >...3.) Related work missing
>
> We have included the related works mentioned by the reviewer and we thank the reviewer for the additional pointers.
> Ying, Rex, et al. "Hierarchical graph representation learning with differentiable pooling." NeurIPS 2018).
> Cao, Defu, et al. "Spectral Temporal Graph Neural Network for Multivariate Time-series Forecasting." NeurIPS 2020.

---

> ### Author Response · Authors · 2021-11-17
> **Response A to Reviewer xjBF**
>
> First of all, thank you for the detailed and extensive feedback, and for expressing that we proposed an interesting approach that follows a promising direction. We answer the raised issues below.
>
> All modifications that we mention throughout this response will be uploaded in a newer version of the paper in the next few days before the rebuttal period ends.
>
> >1.a) For the auxiliary nodes, U, it says they are initialized with random noise, but nothing else is said about them.
>
> The auxiliary nodes U at the input layer are free learnable parameters initialized as random gaussian noise at the beginning of the training and optimized through the training process. We have included a more detailed explanation of this in the paper.
>
> >1.b) The description is a little confusing in a few places. In particular, in Equation 3 and later references back to it).
>
> We wanted to avoid repeating the same equation over the paper. But, we agree it can be clearer to explicitly write down equation 3 when explaining the BP-GNN method instead of referencing eq. 3. We have written down the BP-GNN equations in its full form in the Appendix without referencing Equation 3.
>
> >...1.b) It would be helpful to provide the actual definitions of the various phi functions (at least in the appendix) and dimensions / example dimensions of various inputs and outputs such as the embeddings.
>
> Definitions of the various phi functions are provided in the Appendix, Section A.1. This is mentioned and referenced in the main paper (Method 4, “Architecture details”). Taking your feedback, we have explained them in more detail.
>
> >c.) It would be helpful to understand the actual training process, even if in the appendix or said briefly.
>
> We used the exact same training process as [1] for Experiment 5.1, and [2] for Experiment 5.2. Even if this is mentioned in the paper, and all hyperparameters are provided, we have now explicitly explained how the training batches are sampled from the dataset in training and evaluation to make this clearer as suggested by the reviewer.
>
> [1] Shang, C., Chen, J., and Bi, J. (2021). Discrete graph structure learning for forecasting multiple time series.
> [2] Wu, Z., Pan, S., Long, G., Jiang, J., Chang, X., and Zhang, C. (2020). Connecting the dots: Multivariate time series forecasting with graph neural networks.
>
> >d.) Earlier when describing past work, context c is mentioned. How is context c incorporated into the proposed model, as it seems absent from the method description?
>
> All datasets in the performed experiments didn’t require a context “c”, therefore we didn’t include it in the description of the method. Nevertheless, we could consider including a context “c” in our method by concatenating it in the node embeddings as we did with the unique identifier. Based on your feedback, we have updated the method section of the paper describing this.
>
> >2.a) Various missing experiment details: i. When describing the data, what is "Enc. Length" and "Dec. Length" in Table 2? Is it "Enc. Length" the history window size used in each case, and "Dec. Length" the prediction window size?
>
> Unfortunately, there is a confusion in the literature (as the reviewer points out), where different works use different terms. We have now described in detail what the picked terminology means regardless of which one we choose. In brevity, this is how the terminology comes together:
> Enc. length → Context length → History window size.
> Dec. Length → Prediction length → Horizon length → Prediction window size
>
> >...2.a) For the second set of datasets, why not evaluate predicting multiple time points (such as next 48 hours) as is commonly done in other past work, as opposed to single time point prediction?
>
> We agree with the reviewer that multi-step forecasts are a meaningful experiment that many other forecasting papers rely on. We have a multi-step experiment in Experiment 5.1. To the concrete point of the reviewer, in Experiment 5.2, we actually evaluated in a single time point prediction because this is the setting used in the previous methods included in the table of the experiment. In this experiment, we actually used the exact same training process used in previous works and we just replaced their method with ours to allow for comparability. But, we agree with the reviewer that it would be more meaningful forecasting multiple time points. Unfortunately, in our case, that would require to re-evaluate all previous methods and not only ours.
>
> >2.a.ii. It would also be best to report the periodicity of the time series - i.e., is it daily, hourly, etc…
>
> We have now reported the periodicity of the datasets in the Appendix. For this we have included a new section in the Appendix describing the datasets in more detail.

---

### Decision · Program_Chairs · 2022-01-20

**Decision:**

Reject

**Comment:**

This paper studies multivariate time series forecasting by making relational inference in a latent space. It attempts to address the important issue of reducing the computational complexity of the inferred graph. This motivation is well articulated.

Despite its merits, concerns have been raised regarding the relatively weak evaluation without using datasets involving more many nodes to demonstrate the scalability of the proposed method, which is a major selling point of the paper. As such, while the motivation of the work is clear, its experimental evaluation is not thorough enough to demonstrate the scalability of the proposed method.

The authors made the remark in their response that they are not aware of any public time series dataset of this size (which is not agreed by another reviewer who pointed out that some much larger datasets were used in other papers). Note that it is not uncommon in other work to use synthetic datasets to evaluate the scalability as well as other properties of the proposed methods.

Moreover, clarity of the presentation also has room for improvement.

The paper has potential for publication in a top venue if the comments and suggestions are incorporated to revise the paper.